# Association of Toll-Like Receptor Gene Polymorphisms with Tuberculosis in HIV-Positive Participants

Svetlana Salamaikina [1,*], Vitaly Korchagin [1], Ekaterina Kulabukhova [1,2], Konstantin Mironov [1], Vera Zimina [2], Alexey Kravtchenko [1] and Vasily Akimkin [1]

[1] Central Research Institute of Epidemiology Federal Service for Surveillance on Consumer Rights Protection and Human Wellbeing Russian Federation, Novogireevskaya Str. 3a, 111123 Moscow, Russia

[2] Medical Institute, The Peoples' Friendship University of Russia (RUDN University), Miklukho-Maklaya Str. 6, 117198 Moscow, Russia

[*] Correspondence: salamaikina.sa@phystech.edu

**Abstract:** Genetic factors in the HIV-background may play a significant role in the susceptibility to secondary diseases, like tuberculosis, which is the leading cause in mortality of HIV-positive people. Toll-like receptors (TLRs) are considered to be receptors for adaptive immunity, and polymorphisms in TLR genes can influence the activity of the immune response to infection. We conducted a case–control study of the association of TLR gene polymorphisms with the risk of tuberculosis coinfection in a multi-country sample of HIV-positive participants. Our study revealed certain associations between *TLR4* and *TLR6* polymorphisms and HIV–tuberculosis coinfection. We also found that the analyzed *TLR1* and *TLR4* polymorphisms were linked with the decline in CD4+ cell count, which is a predictor of disease progression in HIV-infected individuals. Our findings confirm that TLR gene polymorphisms are factors that may contribute to development of HIV–tuberculosis coinfection. However, the essence of the observed associations remains unclear, since it can also include both environmental factors and epigenetic mechanisms of gene expression regulation.

**Keywords:** TLR; SNP; CD4 cells; infectious disease; genetic susceptibility; HIV; tuberculosis

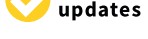



## 1. Introduction

Worldwide, 187,000 HIV-positive patients died of tuberculosis in 2021. HIV-positive individuals are at a higher risk of developing tuberculosis than the general population. In a number of EECA countries, the situation remains unfavorable with regard to the incidence and prevalence of HIV–tuberculosis coinfection [1]. The clinical features of pulmonary tuberculosis in HIV infection are characterized by an atypical clinical presentation and rapid progression, which makes it difficult to diagnose the disease in time. Antiretroviral therapy and chemoprophylaxis for tuberculosis are two primary ways to prevent active tuberculosis in patients with HIV infection [2,3]. Modern antiretroviral treatment (ART) regimens can block HIV-1 replication, but they can not suppress virus production by cells with integrated proviruses; thus, the therapy should not be interrupted. Despite the significant impact of ART in suppressing HIV, complete recovery has not been yet achieved, and the virus continues to persist in reservoirs. Meanwhile, several studies have shown that adherence to tuberculosis chemoprophylaxis among patients with HIV infection continues to be low [4]. One way to increase adherence to therapy is through a personalized approach. In addition to socio-demographic factors, genetic predisposition has been proven to be important in the development of tuberculosis.

The efficiency of chemoprophylaxis depends on patient compliance. Unfortunately, not all patients prescribed chemoprophylaxis are fully compliant, which significantly reduces its effectiveness. The individual risk of tuberculosis should be determined before chemoprophylaxis is prescribed. An individualized approach may include additional measures to increase adherence to TB chemoprophylaxis in high-risk patients: social

support during the period of chemoprophylaxis, more detailed counseling of patients about the benefits of TB chemoprophylaxis [5], and the use of short TB chemoprophylaxis regimens [6,7]. In addition, the monitoring and management of adverse events and the selection of TB chemoprophylaxis regimens for patients with HIV infection are important, taking into account drug–drug interactions [8,9].

According to our previous study, the group of patients coinfected with HIV and tuberculosis is the most disadvantaged in terms of socioeconomic risk factors and tobacco smoking. There is a large proportion of patients with advanced HIV infection alongside severe immunosuppression. At the same time, certain countries in our study have a low coverage of preventive treatment of tuberculosis, despite the high risk of active disease due to the presence of a combination of risk factors.

One of the important parts of the immune system are Toll-like receptors (TLRs), which are responsible for pathogenetic pattern recognition. There are ten types of TLRs (TLR1–TLR10) identified in humans, mostly located on the surface of immune cells, which allows them to recognize a pathogen that has not entered the cell before and activate the MyD88 signaling pathway. When Toll-like receptors (TLRs) are activated and assembled on the membrane, MyD88- or TRIF-adapter proteins are recruited to the cytoplasmic side to form "middosomes" or "trifosomes", respectively. These supramolecular pathways direct signaling events that activate the MAPK and NF-κB pathways, triggering the cell's inflammatory response. This response includes the activation of co-stimulatory molecules and antigen presentation by MHC molecules, as well as the secretion of soluble factors such as cytokines in macrophages and dendritic cells (DC) [10].

The interactions between *M. tuberculosis* and HIV-infected immune cells are still widely unexplored. The in vitro activation of CD4+ T cells through TLR agonists with increased expression of chemokines and antiviral factors increases susceptibility to HIV infection [11]. HIV persists as a provirus in long-lived CD4+ T cells. Recent data indicate that resting CD4+ T cells are one of reservoirs of the virus and a primary barrier to HIV-1 elimination, despite ART [12]. The latent virus resumes replication [13]; nevertheless, there is still a lack of understanding regarding the CD4+ T-cell subsets or regions of the genome acting as reservoirs for the latent HIV provirus. HIV infection inhibits molecular expression (MyD88 and IRAK4) [14], which reduces the production of cytokines and chemokines required to control the spread of TB in HIV-infected individuals. Over the past 20 years, several research works have provided support for using Toll-like receptor agonists in HIV-1 treatment strategies [15]. TLR signals can independently reactivate HIV latency in certain cellular models of HIV latency [16]. Thus, researchers have reported some controversial results from in vivo studies regarding the possible reversal of latency using TLR agonists [17].

Single-nucleotide polymorphisms in TLR genes have been confirmed as related to both HIV and to the recognition of lipopolysaccharides on the surface of mycobacteria that cause tuberculosis [18]. However, the current data on the associations of HIV–tuberculosis coinfection with SNPs in TLR genes are contradictory [19–22]. HIV acts as a factor interacting with distinct host genetic components and affecting the susceptibility to tuberculosis [23]. The *TLR4* D299G polymorphism (rs4986790) has been associated with a risk factor for tuberculosis in HIV-infected patients in Tanzania and Spain, despite no correlation in non-HIV tuberculosis patients [24,25]. TLR8 agonists inhibit HIV infection via a Vit D- and CAMP-dependent autophagy mechanism in human macrophages [26]. The HIV-1 structural proteins p17, p24, and gp41 act as ligands for TLR2 [27]. *TLR2* -975C/T was protective against tuberculosis, due to the attenuation of TLR2 signaling [28]. TLR4 is activated by bacterial lipopolysaccharides, as well as the synthetic compound monophosphoryl lipid A (MPLA) [29]. The *TLR4* rs4986790 polymorphism is a risk factor for active tuberculosis in Caucasian HIV-infected patients [25]. Shi et al. have shown that the *TLR7* rs179008 and *TLR9* rs352140 variants influence the risk of HIV infection [30]. TLR3, TLR7, and TLR9 receptor activity provides cellular immunity in acute infection but accelerates the course of disease in chronic HIV infection with *M. tuberculosis* coinfection [31]. LPS activation by

TLR4 stimulates the programmed death of central and effector memory CD4+ T cells, but a distinct association of these findings with the D299G polymorphism is still unclear [32]. The *TLR8* A1G SNP (rs3764880: M > V) repositions the starting codon from the first exon to the second, leading to an N-truncated variant depleted of three amino acids, which has been reported to be hypermorphic [33]. This SNP has been significantly associated with tuberculosis in Indonesian, Russian, and Moldavian subjects, in Turkish children, and in two Indian cohorts for both pulmonary tuberculosis and extrapulmonary tuberculosis. As shown in several studies, the *TLR8* G allele (rs3764880A/G) was associated with susceptibility to tuberculosis and bacterial load; the *TLR2* CC genotype (rs3804100C/T) was associated with susceptibility to latent tuberculosis infection; and the *TLR4* SNP rs4986790A/G (Asp299Gly) was associated with susceptibility to pulmonary tuberculosis in the Iranian population [34–41]. The key receptors for *M. tuberculosis* recognition are TLR1, TLR2, and TLR6 [42]. SNPs of TLR genes have been associated with an increased risk of *M. tuberculosis* infection or disease [36,43,44].

Recently, it has been proposed that epigenetics may play a role in gene–environment interaction effects contributing to the development of autoimmune and infectious diseases [45,46]. It has been shown that there is an adaptive element to the innate immune response, and that this adaptation is based on epigenetic mechanisms [47]. It could be possible that epigenetic mechanisms affect the TLR system through changes in TLR gene expression. Regardless, the role of epigenetics in this is not well studied.

The aim of our study was to analyze the genetic risk factors of HIV–tuberculosis coinfection in Eastern European and Central Asian countries' populations among HIV-positive individuals. Previously, we analyzed the association of SNPs in *TLR2* and *TLR4* genes in samples of HIV patients and HIV–tuberculosis patients in Eastern Europe and Central Asia. In particular, significant associations of SNPs were shown for the Russian population [48]. The results obtained earlier identified associations in individual territories. At the same time, comprehensive analysis of SNP allele frequencies in HIV and HIV–tuberculosis samples has not previously been performed, in particular for *TLR6* and *TLR8* as the most promising markers of the genetic risk of coinfection.

## 2. Results

### *2.1. The Demographic Characteristics of the Samples under Study*

The gender and age structure of the samples under study belonging to people living with HIV and coinfected with tuberculosis is presented in Table 1.

**Table 1.** Demographic characteristics of study samples.

|  | Case, n (%) | Control, n (%) | Pearson's Chi-Squared Test, $X^2$ (*p*) |
|---|---|---|---|
| Gender: |  |  |  |
| Male | 179 (71.6) | 174 (69.6) | 0.24 (0.62) |
| Female | 71 (28.4) | 76 (30.4) |  |
| Age: |  |  |  |
| Under 35 | 55 (22) | 55 (22) |  |
| 35–44 | 110 (44) | 107 (42.8) | 0.09 (0.95) |
| 44 and older | 85 (34) | 88 (35.2) |  |

All patients in both groups were receiving ART at the time of enrollment in the study.

### *2.2. TLR Genes Polymorphisms Association with Tuberculosis Coinfection*

Six candidate SNPs in five TLR genes have been analyzed: rs5743551 (*TLR1*), rs5743708 (*TLR2*), rs3804100 (*TLR2*), rs4986790 (*TLR4*), rs5743810 (*TLR6*), and rs3764880 (*TLR8*). Genotypic frequency distributions were in accordance with the Hardy–Weinberg equilibrium in both HIV and HIV–tuberculosis samples for all loci except rs5743810 (*TLR6*) in HIV patients (*p* = 0.036).

We did not observe any significant association of tuberculosis coinfection in HIV patients with any of the analyzed SNPs after multiple test correction. Estimates for the

models for each SNP are presented in Supplementary Table S2. However, a trend was observed for the rs4986790 (*TLR4*) and rs5743810 (*TLR6*) gene variants. After allele grouping to evaluate different inheritance models, we found a contribution of the rs4986790-G (*TLR4*) allele according to the dominant model (OR = 0.50) and the rs5743810-A (*TLR6*) allele according to the recessive model (OR = 0.38) (Table 2). In both cases, the frequency of the rare allele in the case group was lower than in the control group. Thus, rare alleles were protective in relation to the risk of developing tuberculosis coinfection against the background of HIV.

**Table 2.** Association of *TLR4* and *TLR6* genotypes with the risk of tuberculosis coinfection in HIV-positive individuals.

| SNP | Control, n (%) | Case, n (%) | OR (CI 95%) | $p$ ($p_{FDR\text{-}BH}$) |
|---|---|---|---|---|
| rs4986790 (*TLR4*) | | | | |
| A/A | 207 (84.1) | 223 (91.4) | 1.00 | 0.014 (0.067) |
| A/G-G/G | 39 (15.9) | 21 (8.6) | 0.50 (0.28–0.88) | |
| rs5743810 (*TLR6*) | | | | |
| G/G-A/G | 226 (91.9) | 236 (96.7) | 1.00 | 0.019 (0.067) |
| A/A | 20 (8.1) | 8 (3.3) | 0.38 (0.17–0.89) | |

*2.3. TLR Gene Polymorphisms' Association with Changes in CD4+ Cell Count in HIV-Positive Individuals*

Assuming that genetic factors may be associated with the risk of tuberculosis coinfection indirectly, through linked factors or mechanisms, we analyzed the association of TLR gene polymorphisms with the level of the CD4+ cell count, as one of the crucial factors in the development of opportunistic infections in patients with HIV. The concentration of 200 cells per $mm^3$ was taken as the threshold [49,50]. For further analysis, all patients were redistributed into groups according to the CD4+ cell count at the onset of tuberculosis development. Thus, patients with a CD4+ cell count of less than 200 cells/$mm^3$ formed the "CD4 low" group, while those with a CD4+ cell count greater than or equal to 200 cells/$mm^3$ formed the "CD4 high" group. After adjustment for tuberculosis status, the rs4986790-G (*TLR4*) and rs5743551-G (*TLR1*) alleles were significantly associated with the changes in CD4+ cell count above/below 200 cells/$mm^3$. Thus, the rs4986790-G (*TLR4*) allele was protective against CD4+ cell count reduction (OR = 0.28), and the rs5743551-G (*TLR1*) allele was a risk allele (OR = 2.16). These associations still existed after FDR-BH correction ($p < 0.05$) (Table 3). There was no statistically significant association of the rs5743810 (*TLR6*) polymorphism with the changes in CD4+ cell count after multiple test correction.

**Table 3.** Association of TLR gene polymorphisms with crucial changes in CD4+ cell count in patients with HIV.

| SNP | «CD4 High», n (%) | «CD4 Low», n (%) | OR (CI 95%) | $p$ ($p_{FDR\text{-}BH}$) | $p_{FDR\text{-}BH}$ Adjusted by Tuberculosis Coinfection |
|---|---|---|---|---|---|
| rs4986790 (*TLR4*) | | | | | |
| A/A | 265 (83.9) | 165 (94.8) | 1.00 | 0.00017 (0.00067) | 0.0044 |
| A/G-G/G | 51 (16.1) | 9 (5.2) | 0.28 (0.14–0.59) | | |
| rs5743551 (*TLR1*) | | | | | |
| A/A | 158 (50.0) | 55 (31.6) | 1.00 | $7.35 \times 10^{-5}$ (0.00059) | 0.0013 |
| A/G-G/G | 158 (50.0) | 119 (68.4) | 2.16 (1.47–3.19) | | |
| rs5743810 (*TLR6*) | | | | | |
| G/G-A/G | 293 (92.7) | 169 (97.1) | 1.00 | 0.034 (0.068) | 0.2432 |
| A/A | 23 (7.3) | 5 (2.9) | 0.38 (0.14–1.01) | | |

Next, we analyzed differences in CD4+ cell counts between carriers of different genotypes of two SNPs: rs4986790 (*TLR4*) and rs5743551 (*TLR1*). In the entire sample of patients, the CD4+ cell median count was 334 cells/mm$^3$ [Q1–Q3: 131.2–570.2]. Patients with rs4986790-AG/GG (*TLR4*) genotypes had significantly higher CD4+ cell counts (521 [316.2–641.2] cells/mm$^3$) than patients with the rs4986790-AA (*TLR4*) genotype (300 [120–553.2] cells/mm$^3$), $p < 0.05$ (Figure 1a). On the contrary, patients with rs5743551-AG/GG (*TLR1*) genotypes had a significantly lower CD4+ cell count (273 [115–540] cells/mm$^3$) than patients with the rs5743551-AA (*TLR1*) genotype (397 [193–605] cells/mm$^3$) (Figure 1b).

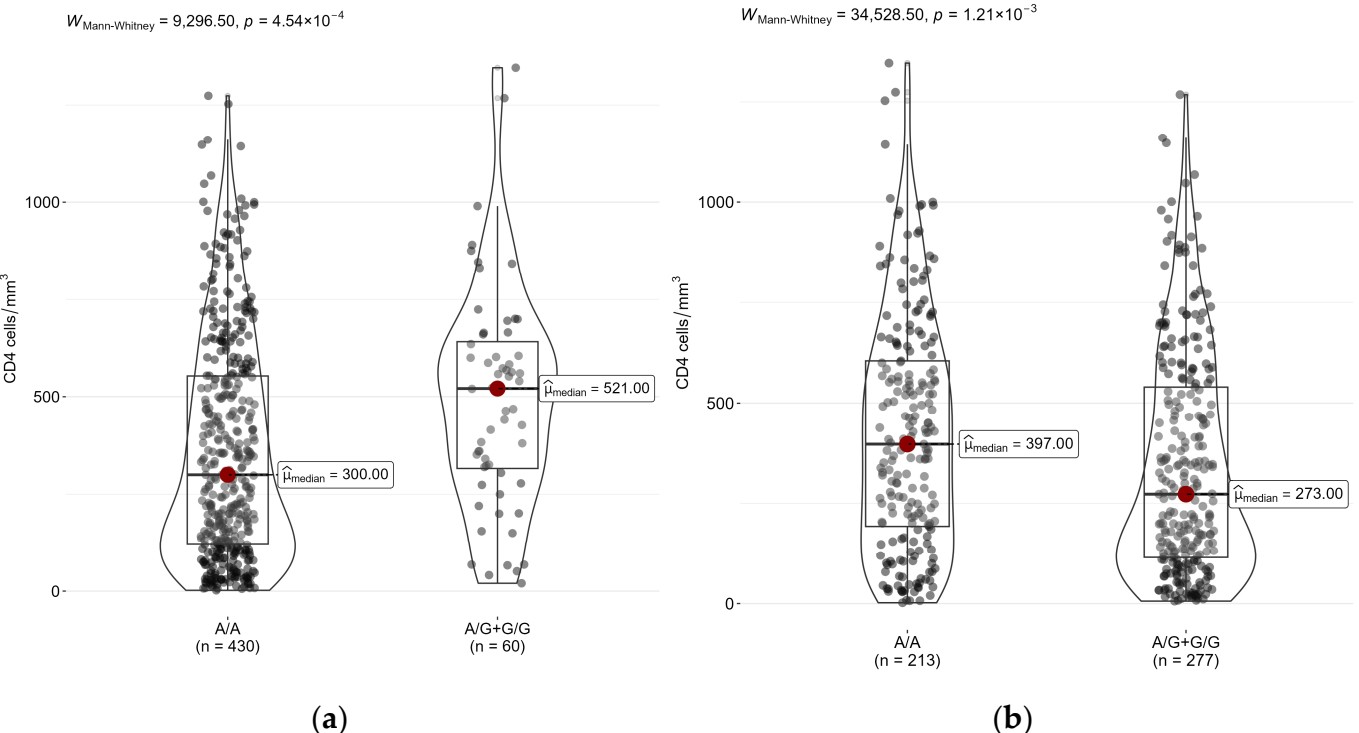

**Figure 1.** Differences in CD4+ cell counts between genotypes of (**a**) rs4986790 (*TLR4*) and (**b**) rs5743551 (*TLR1*) in HIV patients.

As a result of the statistically ($p < 0.0001$) and clinically significant differences in CD4+ cell count between HIV–tuberculosis coinfected patients (median conc. 155 [68–336.5] cells/mm$^3$) and HIV patients (median conc. 522 [326.2–700] cells/mm$^3$), we analyzed each group separately. The previous trend was observed in both samples.

The median level of the CD4+ cell count in the rs4986790-AA (*TLR4*) genotype carriers was 1.13/2.23 times lower than in the rs4986790-AG/GG (*TLR4*) genotype carriers in HIV patients and HIV–tuberculosis coinfected patients, respectively (Figure 2a). The median level of the CD4+ cell count in the rs5743551-AA (*TLR1*) genotype carriers compared to the rs5743551-AG/GG (*TLR1*) genotype carriers was 1.08 times higher in the control group and 1.28 times higher in the case group (Figure 2b). However, the observed differences reached statistical significance ($p < 0.05$) only for the rs4986790 (*TLR4*) polymorphism in the case group.

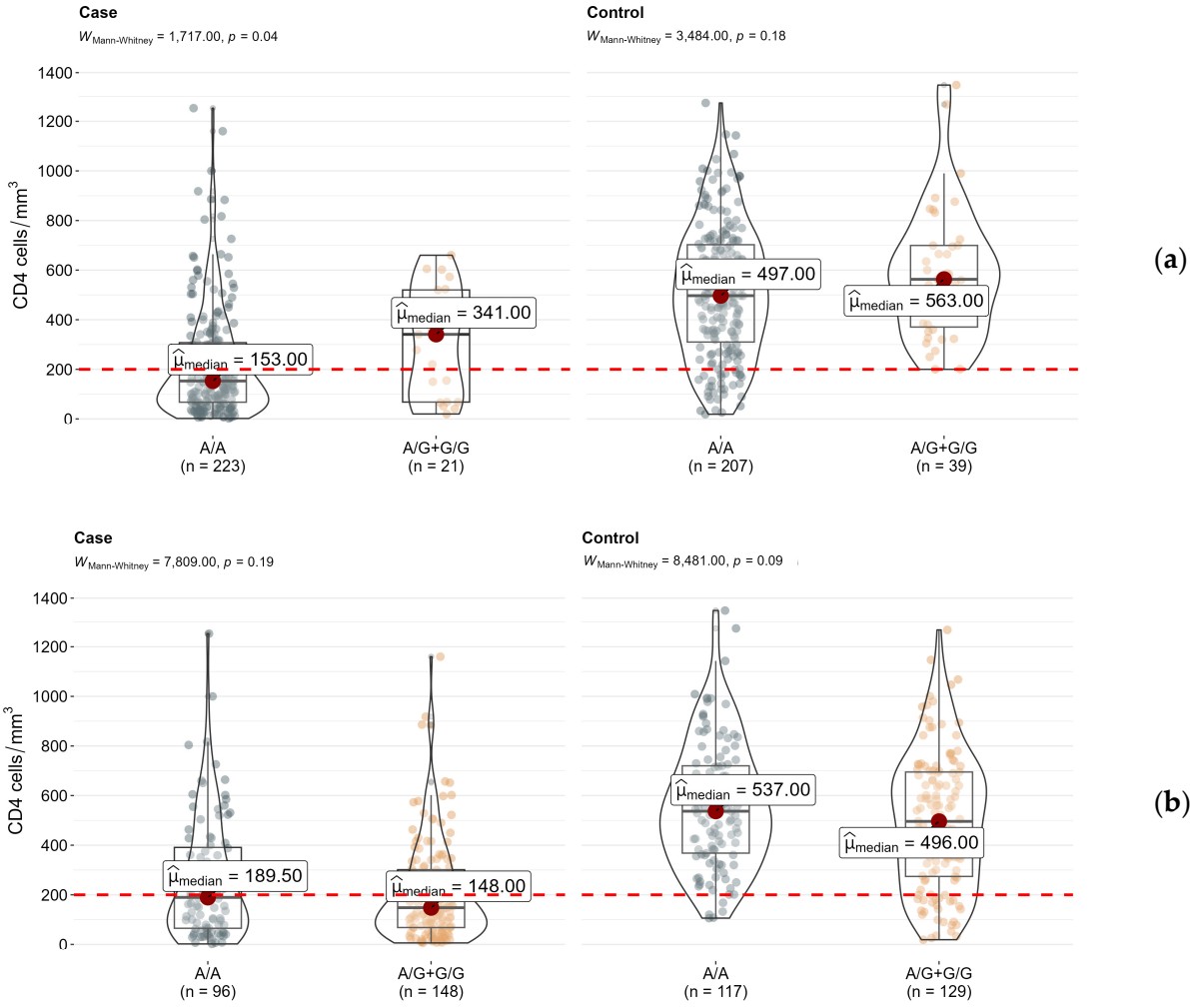

**Figure 2.** Differences in CD4+ cell counts between genotypes of (**a**) rs4986790 (*TLR4*) and (**b**) rs5743551 (*TLR1*) in subsamples of HIV (control) and HIV–tuberculosis coinfected patients (case).

## 3. Discussion

While TLRs are classified as part of the nonspecific immune response, their influence extends to the development of specific immunity, including the differentiation of T-helper cells. Consequently, changes in TLR expression can lead to alterations in the count of CD4+ lymphocytes, which are of chief importance in the pathogenesis of HIV infection and tuberculosis, affecting the risk of developing these diseases. TLR2 has been shown to synergize with IL-4 and TGF-β at the signaling level, inducing TH9 differentiation [51]. However, the complex interactions between TLRs and the immune system are not fully understood, despite intensive studies over the past decade, and our research only provides a superficial understanding of these mechanisms, making it difficult to draw definitive conclusions based on the data obtained.

It is shown that TLR expression is elevated in innate immune cells in patients with a high HIV-1 burden and coinfection with opportunistic pathogens. Hernández et. al hypothesized that the modulation of TLR expression is a mechanism that contributes to HIV-1 replication and AIDS pathogenesis in patients coinfected with opportunistic pathogens [52].

Insights on mechanisms by which HIV infection inhibits the expression of molecules involved in TLR2-mediated pathways will facilitate the development of modern immunomodulatory strategies of tuberculosis in HIV-positive individuals to enhance innate immunity to *M. tuberculosis* and prevent latent tuberculosis infection progression to tuberculosis in the highly susceptible HIV-infected population [14].

The occurrence of comorbid diseases against the background of HIV is influenced not only by environmental factors but also by the level of CD4+ cell suppression by HIV. Despite taking antiretroviral therapy (ART), certain people living with HIV have low CD4+ cell counts. We hypothesized that low CD4+ cell counts in people living with HIV and taking ART may be related to genetic variants of TLR genes.

Our results describes the frequency distribution of rare alleles of TLR gene polymorphisms and describes their association with pulmonary tuberculosis infection in people living with HIV. Studies of the genetic factors involved in complex diseases have not yet provided clear explanations for the onset of such diseases, although they may help to identify the risk factors for contracting the infection.

Our findings demonstrate a correlation between the genotype and the rate of CD4+ cell count decline, which increases the risk of developing coexisting diseases beyond tuberculosis. TLRs located on the surface of immune cells have been shown to recognize HIV-1 [53,54]. However, the results obtained in Section 2.1 could be affected by other factors not considered in our analysis, which is one of the limitations of our study. While a decrease in CD4+ cell count to less than 200 cells/mm$^3$ is a direct and immediate risk factor that is independent of nationality and country of residence, social factors may vary significantly between countries and impact immunity indirectly.

The rs5743551 and rs4986790 polymorphisms can affect the level of expression of the *TLR1* and *TLR4* genes. However, such associations have not previously been shown against the background of infectious diseases. The expression of TLR genes may be associated with susceptibility to pathogens, especially in the form of the immunosuppression caused by HIV [55].

Certain studies have indicated that rs4986790 (*TLR4*) is associated with a change in the extracellular domain structure of the TLR4 receptor, suggesting that the allele G is associated with an attenuated immune response to LPS and lower secretion levels of pro-inflammatory cytokines [56–58]. Our previous study demonstrated that the rs4986790-G (*TLR4*) allele is a risk factor, but increased sampling and analysis of several populations did not confirm the result [45].

A detailed description of the clusters formed by the *TLR1*, *TLR2*, *TLR6*, and *TLR10* genes is described in a paper by Henrick et al. It is likely that the association of rs5743810 (*TLR6*) observed by us may be related to the sensitivity of TLR family receptors to HIV proteins [59].

Najwa A Mhmoud's results published in 2023 showed that the *TLR8* rs3764879 and rs3764880 alleles were more frequent in the patient population compared to the healthy control population [60]. Our results showed no statistically significant differences in the frequency of the rare allele among HIV and HIV–tuberculosis groups.

The restoration of CD4+ T cells in HIV-infected individuals who receive antiretroviral therapy (ART) has been shown to be associated with specific genotypes that are linked to lipopolysaccharide (LPS) metabolism [61]. The frequencies of SNPs in the TLR genes compared during case–control studies may indicate the genetic characteristics of groups of people who develop infectious diseases. The information we obtained about the protective effect of the rs4986790-G (*TLR4*) allele in comparison with information about the level of TLR gene expression in tuberculosis-coinfected people living with HIV will allow us to determine more clearly the influence of genetic factors in the development of infectious diseases.

The study identified the innate immune factors associated with tuberculosis recurrence in individuals treated with ART for HIV infection, who had a previous history of successful tuberculosis treatment [62]. The complex effects of HIV-1 and ART on cell-mediated immunity can potentially compromise the body's ability to mount an effective defense against tuberculosis, thereby offering novel insights into the immune correlates of defense and pathogenesis [63].

HIV infection results in a reduction in the count of CD4+ lymphocytes, and once this number drops below a certain threshold, patients become vulnerable to secondary infec-

tions. Tuberculosis is one such opportunistic infection that affects HIV-infected individuals, and the level of CD4+ cell count can be considered as a risk factor for the development of tuberculosis in this population. According to the data presented in this study, CD4+ T-cell counts were lower in people living with HIV and TB than in people not infected with *M. tuberculosis*.

In connection with the fact that CD4+ T cells act as reservoirs for latent HIV, infection with *M. tuberculosis* induces HIV to exit the reservoirs and reduces CD4+ cell counts. This mechanism might rely on activation by TLR receptors in downstream cascades such as MyD88 [64]. It has been shown that even short-term infection with microbial pathogens can promote TLR2 activation and presumably contribute to a short-term increase in viral load in patients with HIV [65]. However, the mechanism of such reactivation is not fully understood. In myeloid monocytic suppressor cells infected with HIV, rapid signal transduction through the TLR–MyD88 mechanism contributes to an increased mycobacterial load and consequently increases the risk of developing active tuberculosis during viral suppression [66].

The associations we observed may be attributed to other functional SNPs that are in linkage disequilibrium with the SNPs we studied. Various other factors also influence the risk of developing tuberculosis. For instance, the specifics of the organization and screening for latent tuberculosis infection and the extent of its treatment among HIV-positive patients in the region, timely initiation of antiretroviral therapy, and coverage, as well as social and economic factors. Therefore, additional rigorous work is necessary, which could be conducted in the future.

One potential risk factor for tuberculosis is the epigenetic regulation of gene expression. Chen et al. have shown that the methylation of certain CpG sites in the TLR2 promoter reduces TLR2 expression levels in natural killer (NK) cells/monocytes from patients with active pulmonary tuberculosis and correlates with bacterial load and disease severity [67]. Thus, DNA hypermethylation in these monocytes is thought to reflect disease severity [68]. These results suggest that DNA methylation in leukocytes can be used by clinical prognostic tools for the chemoprevention and treatment of tuberculosis.

## 4. Materials and Methods

### 4.1. Study Design

Depersonalized samples from HIV patients (control group, n = 250) and HIV–tuberculosis (case group, n = 250) patients were used in the study. The samples were collected during 2019 in the Russian Federation, the Republic of Belarus, the Republic of Armenia, the Kyrgyzstan Republic, and the Republic of Tajikistan.

HIV was diagnosed in accordance with country-specific guidelines based on the WHO guidelines. This diagnosis was made on the basis of two positive ELISA tests confirmed by immunoblot or three positive ELISA tests, depending on the year of establishment [49,50].

The inclusion criteria for HIV patients were the confirmed presence of HIV and data regarding CD4+ cell count and viral load at the time of inclusion in the study (no more than 6 months of follow-up).

The inclusion criteria for patients with coinfection were a tuberculosis case confirmed by isolation of *M. tuberculosis* from biological material by a molecular genetic method (detection of MBT DNA), first-time diagnosed tuberculosis of any localization, developed with the background of HIV, and available data regarding CD4+ cell count and viral load test results at the time of tuberculosis diagnosis (at the earliest 6 months prior to tuberculosis diagnosis and at the latest 2 months after tuberculosis diagnosis).

The exclusion criteria were acute HIV infection, tuberculosis which was diagnosed only on the basis of circumstantial signs (characteristic syndrome, data of radiological examination), without isolation of *M. tuberculosis* from biological material, the presence of a mental disorder in the patient at the time of the study, and being in a state of alcohol or drug intoxication.

Clinical and demographic characteristics have been published previously [69].

*4.2. CD4+ T-Cell Counts, DNA Isolation and Genotyping*

CD4+ T-cell counts were obtained via the flow cytofluorimeter FACSCalibur (Becton Dickinson, USA) according to the standard manufacturer's protocols.

Peripheral venous blood samples anticoagulated by EDTA from every participant were stored at 4 °C. Genomic DNA was isolated from collected blood samples using the RIBO-prep kit (AmpliSens, Moscow, Russia), following the manufacturer's protocol.

Real-time PCR primers and probes were designed for appropriate alleles using reference sequences from the SNP database [70]. We followed the design guidelines according to real-time recommendations from the instrument manufacturer (Qiagen, Hilden, Germany) [71]. The annealing temperature for the probes was adjusted to 60 °C. The primers' and probes' concentrations were chosen empirically and ranged from 0.2 to 0.4 and from 0.06 to 0.12 μM, respectively. The designed oligonucleotide sequences have been presented previously [72]. The SNP-related primers are listed in Supplementary Table S1.

PCR reactions were carried out in a final volume of 25 μL, containing 10 μL of extracted DNA, 10 μL of dNTPs (0.44 mM), primers, probes, 4.5 μL of RT-PCR-mix-2 FEP/FRT reagents, and 0.5 μL of TaqF polymerase. The real-time PCR amplification was performed using the Rotor Gene Q cycler (Qiagen, Hilden, Germany) with the following procedure: 95 °C—15 min (1 cycle); 95 °C—5 s/60 °C—20 s/72 °C—10 s (45 cycles with fluorescent signal detection at 60 °C). In the amplification results analysis, the threshold was set at 10% of the highest fluorescent signal value for each channel.

The confirmation of the real-time PCR results was performed by PyroMark Q24 sequencing using the reagent kits recommended by the equipment manufacturer (Qiagen, Hilden, Germany).

In total, 4 samples from the control group and 6 samples from the case group were not suitable for genotyping. Thus, genotyping results were analyzed for 246 and 244 samples, respectively.

*4.3. Statistical Analysis*

Statistical analysis was performed with the R environment (version 4.2.2) including integrated functions for the analysis of contingency tables (Pearson's χ2 test and Fisher's exact test) and nonparametric analysis of differences between groups (Mann–Whitney U test). Association and risk scores (ORs) were performed using SNPassoc [73] and epitools [74] packages according to prespecified genetic risk models—codominant, dominant, recessive, overdominant, and log-additive, as described earlier [75,76]. The genetic characteristics of the samples were analyzed using the HardyWeinberg [77] packages. The Bonferroni method and Benjamini–Hochberg false discovery rate (FDR-BH approach) were used to adjust for multiple comparisons, where necessary. The results were considered statistically significant at $p < 0.05$. The ethnicities and recruitment sites were combined into one to avoid small sample size issues within each sample. All graphical results were generated using the ggplot2 [78] and ggstatsplot [79] packages.

## 5. Conclusions

According to our findings, patients with the rs4986790-G (*TLR4*) allele had higher concentrations of CD4+ lymphocytes compared to patients without the rs4986790-G (*TLR4*) allele in both the entire sample of HIV patients and in groups with/without tuberculosis coinfection. Moreover, HIV patients with the rs4986790-G (*TLR4*) allele are less likely to develop tuberculosis. Thus, polymorphisms in the TLR genes may indirectly affect the risk of coinfection in patients with HIV (e.g., through effects at the immune cell level).

## 6. Limitations

This study had several limitations. Firstly, it did not take into account additional external factors that may meaningfully contribute to the occurrence of tuberculosis in HIV-infected individuals. Secondly, the analysis of genetic factors was not carried out for all of the study populations across the five countries included in the study. In future

studies, it is recommended to conduct a comprehensive analysis of population risks for each participating country, incorporating the results obtained from this study.

**Supplementary Materials:** The following supporting information can be downloaded at: https://www.mdpi.com/article/10.3390/epigenomes7030015/s1, Table S1: Oligonucleotide characteristics; Table S2: Association of studied TLR gene polymorphisms genotypes with the risk of tuberculosis coinfection in HIV-positive individuals.

**Author Contributions:** Conceptualization, A.K. and V.Z.; methodology, K.M.; validation, S.S.; formal analysis, S.S. and V.K.; investigation, S.S. and E.K.; resources, E.K. and V.Z.; data curation, V.K.; writing—original draft preparation, S.S. and V.K.; writing—review and editing, S.S. and V.K.; visualization, V.K.; supervision, A.K. and K.M.; project administration, V.A.; funding acquisition, V.A. All authors have read and agreed to the published version of the manuscript.

**Funding:** This research received no external funding.

**Institutional Review Board Statement:** This study was conducted in accordance with the Declaration of Helsinki, and approved by the Ethics Committee of the Central Research Institute of Epidemiology, Federal Service for Surveillance on Consumer Rights Protection and Human Wellbeing, Russian Federation (protocol code: 90 26.03.2019).

**Informed Consent Statement:** Informed consent was obtained from all subjects involved in the study.

**Data Availability Statement:** The data that support the findings of this study are available from the corresponding author upon reasonable request.

**Acknowledgments:** The authors acknowledge Olga Dribnokhodova, Elena Pozdysheva, Irina Gaponova, Anna Esman, Anastasia Pokrovskaya, Zoya Suvorova, Olga Khoklova, Grigoriy Volchenkov, Tatyana Zamkovaya, Narina Sarkisyants, Astkhik Davidyan, Siranush Martoyan, Svetlana Sergeenko, Marina Gorovaya, Zhanna Saprykina, Aibek Bekbolotov, Aida Mamyrbayeva, Elmira Abdrahmanova, Safarkhon Sattorov, Alidjon Soliev, Shahnoza Azamova, for data collection, laboratory analysis, management of sample databases, and epidemiological data finalization.

**Conflicts of Interest:** The authors declare no conflict of interest.

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
