# Peer review of "Association of Toll-Like Receptor Gene Polymorphisms with Tuberculosis in HIV-Positive Participants"

_2075-4655, 2023_

Round 1

Reviewer 1 Report

Report on manuscript 2453853

In the manuscript “Association Of Toll-Like Receptors Genes (TLR) Polymorphisms With Tuberculosis In HIV-positive Participants”, Svetlana Salamaikina and colleagues try to understand whether single-nucleotide polymorphisms (SNPs) on Toll-like receptors (TLRs) are associated with an increased incidence of Tuberculosis on HIV-infected patients.

For this, the authors enrol 250 single HIV-infected individuals and 250 HIV and Mycobacterium tuberculosis (Mtb) co-infected individuals. They observe that two SNPs (on TLR4 and TLR6) might be protective for co-infection, as they are under-represented in the co-infected group. Next, they look whether three SNPs on TLRs might be affecting the number of CD4 T cells and observe that an SNP on TLR4 is more prevalent in patients with high number of CD4 T cells. The SNP identified on TLR6 that might be protective for co-infection does not seem to have an effect on the number of CD4 T cells. Interestingly, the authors observe that an SNP on TLR1 might lead to lower number of CD4 T cells and the same SNP shows a tendency to increase the susceptibility to HIV and Mtb co-infection.

The paper is well written, the goal is clear and methods are carefully explained. The authors have a big cohort of patients (more than 244 per group). I have a few suggestions, but nothing that prevents the manuscript from being published.

Major points:

1)      The authors claim they have 250 patients per group in Table 1 but have only genotyping data for 246 and 244 patients (lines 293-295). They should amend Table 1, adding a line with this information.

2)      The authors should show the complete analysis that was performed for the correlation between SNPs of TLRs and CD4 T cell counts, not just for TLR1, TLR4 and TLR6 (Table 3).

3)      The authors should discuss the role of TLR1 SNP that causes a drop in CD4 T cell count and how it might be a risk factor for co-infection, and how might the TLR6 SNP affect the risk for co-infection. The current discussion is lacking entirely on TLR6 and only mentions that the SNP for TLR1 affects its expression (higher expression, lower expression?)

Minor suggestions:

4)      The authors use the nomenclature “Control” and “Case” throughout the manuscript. Yet, since they change the definition of “Control” and “Case” on Table 2, they should consider referring to the groups by more concrete names, like “HIV” and “HIV+Mtb” like they do for Supplementary Table 2 or “CD4 high” and “CD4 low”. This is especially important for Figure 2 where I assume the authors are talking about “HIV” and “HIV+Mtb” as “Control” and “Case”, respectively.

Author Response

Dear reviewer!

Thank you very much for your notes. We've try to make our manuskript more comprehensible.

Here is our answers to your questions:

1)      The authors claim they have 250 patients per group in Table 1 but have only genotyping data for 246 and 244 patients (lines 293-295). They should amend Table 1, adding a line with this information.
Thank you for your notice. We described this difference in Materials and methods section: “Four samples from the control group and six samples from the case group were not suitable for genotyping. Thus, genotyping results were analyzed for 246 and 244 samples, respectively.”

2)      The authors should show the complete analysis that was performed for the correlation between SNPs of TLRs and CD4 T cell counts, not just for TLR1, TLR4 and TLR6 (Table 3).
We’ve complete our Supplementary file by this data, thank you.

3)      The authors should discuss the role of TLR1 SNP that causes a drop in CD4 T cell count and how it might be a risk factor for co-infection, and how might the TLR6 SNP affect the risk for co-infection. The current discussion is lacking entirely on TLR6 and only mentions that the SNP for TLR1 affects its expression (higher expression, lower expression?)
We have made changes to the discussion section, which includes more recent studies and a discussion of our findings related to TLR1 and TLR6. In addition, we have slightly restructured this part of the paper to make it more comprehensible.

 4)      The authors use the nomenclature “Control” and “Case” throughout the manuscript. Yet, since they change the definition of “Control” and “Case” on Table 2, they should consider referring to the groups by more concrete names, like “HIV” and “HIV+Mtb” like they do for Supplementary Table 2 or “CD4 high” and “CD4 low”. This is especially important for Figure 2 where I assume the authors are talking about “HIV” and “HIV+Mtb” as “Control” and “Case”, respectively.
Thanks for the comment. We have tried to avoid ambiguity in the definition of Case and Control groups. But unfortunately, this led to confusion when we needed introducing additional groups criteria. Since initially in the study the HIV group was defined as Control, and HIV-tuberculosis - as Case, we decided to leave this division as the main one, and only in section 2.3. took your advice and defined the groups in the text and in Table 3 as "CD4 low" and "CD4 high".

The improved version of the manuskript is in the attachment.

Reviewer 2 Report

1.    You might want to specify the title to certain types of TLR’s

2.    Comparing and connecting socioeconomic-demographic/environmental factors to genetic predisposition would make a better impact for this study.

3.    I noticed that you did not study TLR 3, 7,9 polymorphisms despite previous reports, can you please explain your decision

4.    You studied different population in this article, were the results different from your previous study

5.    You mentioned that the rare allele in TLR 4 and 6 were protective against the risk of developing tuberculosis coinfection in HIV, can you put a plausible mechanism for your findings.

6.    You should concentrate your discussion on your specific findings (TLR 4 AND 6 polymorphism) and the relation to TLR1 and 4 polymorphism and CD4 count in the HIV/TB coinfection

7.    Please state abbreviations in full when they are first mentioned in text, for example “HWE”

Author Response

Dear reviewer!

Thank you for your suggestions. Here we've try to answer your questions:

1.    You might want to specify the title to certain types of TLR’s
Thank You for propose. But we carefully considered several title variants and сhoose the latter, as the most concise, but reflecting the essence of the study.

2.    Comparing and connecting socioeconomic-demographic/environmental factors to genetic predisposition would make a better impact for this study.
Thank you for the suggestion. Undoubtedly, including main socioeconomic-demographic/environmental factors to analysis could give us new and important results. But such study would also require a significant increase in the sample. This may be the goal of future studies.

3.    I noticed that you did not study TLR 3, 7,9 polymorphisms despite previous reports, can you please explain your decision
Thank you for your question. TLR3, 7 and 9 are intracellular receptors. Our study was focused only on extracellular TLR gene polymorphisms. 

4.    You studied different population in this article, were the results different from your previous study.
Thank you for the question. Our previous study (Kulabukhova, E.I.; Mironov, K.O.; Dunaeva, E.A.; Kireev, D.E.; Narkevich, A.N.; Zimina, V.N.; Kravchenko, A.V. The association between genetic polymorphisms of toll-like receptors and mannose-binding lectin and active tuberculosis in hiv-infected patients. VIČ-infekc. immunosupr. 2020, 11, 61–69, doi:10.22328/2077-9828-2019-11-4-61-69.) demonstrated rs4986790-G (TLR4) allele as a risk factor, but increased sampling and analysis of several populations at present study did not confirm the result.  We’ve complete the discussion section by this data.

5.    You mentioned that the rare allele in TLR 4 and 6 were protective against the risk of developing tuberculosis coinfection in HIV, can you put a plausible mechanism for your findings.
We suppose that rare alleles of the polymorphisms of TLR4 and TLR6 genes could affect the expression of these genes. Accordingly, the overexpression may affect pathogen recognition by TLR4 and TLR6 receptors and activate the defense mechanism of the cell. We hope that the discussion section became clearer after our revisions.

6.    You should concentrate your discussion on your specific findings (TLR 4 AND 6 polymorphism) and the relation to TLR1 and 4 polymorphism and CD4 count in the HIV/TB coinfection.
Thank You. Following your suggestion, we have updated the discussion section to include more recent research and a discussion of our findings related to TLR1 and TLR6. In addition, we have slightly restructured this part of the manuscript to make it more comprehensible.

7.    Please state abbreviations in full when they are first mentioned in text, for example “HWE”
Thank you. We’ve revised this.

The revised version of the manuskript is in the attachment.

Reviewer 3 Report

Comments for the Authors

epigenomes-2453853

Association Of Toll-Like Receptors Genes (TLR) Polymorphisms With Tuberculosis In HIV-positive Participants

Authors: Svetlana Salamaikina, Vitaly Korchagin, Ekaterina Kulabukhova, Konstantin Mironov, Vera Zimina, Alexey Kravtchenko, Vasily Akimkin

The central theme of this study attempt to correlate the probability of the risk of receiving opportunistic infectious disease, tuberculosis to Toll-like receptors (TLR4 and 6) polymorphisms (SNPs) appearing in HIV-1-infected individuals. 250 HIV-1-infected individuals (control group) and 250 HIV-tuberculosis-coinfected individuals (case group) were enrolled in this study. Salamaikina and colleagues performed SNP genotyping using PCR-based methods on DNA samples isolated from both groups (control v.s. case). The authors claim that although no significant association of tuberculosis coinfection in HIV-1-infected individuals was observed, the frequency of appearance of rs4986790-G (TLR4) and rs5743810-A (TLR6) was relatively low in the case group compared to the control group (Table 2). In addition, Salamaikina and colleagues also seek the correlation between the level of CD4 T-cell count and SNPs shown on LTRs. The authors observe that the rs4986790-G (TLR4) and rs5743551-G (TLR1) were frequently detected in the HIV-1-infected individuals whose CD4 T cells count is less than 200 cells per mm3.

The whole manuscript is more descriptive with insufficient evidence supporting the authors’ hypothesis (lines 184 and 185). In addition, careful proofreading should be required before submitting the manuscript.

Major comments:

1. Line 33: it is debated whether current ART can be considered “effective” in blocking HIV-1 replication. Indeed, current antiretroviral drugs can block new infections of susceptible cells, but not suppress virus production by cells with integrated proviruses, thus the therapy cannot be interrupted. To be honest, none of the currently available antiretroviral strategies can completely purge HIV-1 infections.

2. Line 39: Please describe in detail what the personalized approach is and cite the corresponding references.

3. Line 53: The term “reactivation” is perhaps not a proper word here in this sentence. I went through the reference of Cromarty et al. (2019) [6]. CD4+ T cells that are viral-free should be activated by TLR agonists rather than reactivated.

4. Lines 56-57: Siliciano et al. (2014) [7] is indeed one of the most important milestone papers regarding HIV-1 latency and latent reservoir. However, a lot of progress has been made in the field of HIV-1 latency associated with epigenetic features in the latest 20 years. The authors should provide more updated knowledge at this stage.

5. Line 56: The same as the previous comment, the sentence “.. resting CD4+ T cells are a basic reservoir of …” is too general. What do the authors mean by “basic reservoir”? In addition, the barrier to curing HIV-1 is due to the persistence of (deeply) latent proviruses in reservoir cells. Resting CD4+ T cells are one type of cells where latent proviruses preferentially integrate; however, those cells are not the reason why we cannot completely purge HIV-1 infection. The key is that we still do not know the location of latent reservoir (what subsets of CD4+ T cells or in which genomic regions for example) harboring (deeply) latent proviruses.

6. Lines 59-62: Reference is required for the sentence “Single nucleotide polymorphisms in TLR genes … on the surface of mycobacteria that cause tuberculosis.”.

7. Lines 62-63: Reference is required for the sentence “However, the current data on … with SNPs in TLR genes is contradictory.”.

8. The term “long-term HIV” is ambiguous to me. Is it referred to HIV-1-infected individuals subjected to a long period of ART or a persistent HIV infection?

9. Section 2.1: If applicable, the authors should provide more information about the treatment history of HIV-1-infected individuals, including the period of ART and what kinds of antiretroviral drugs are used. Can different combinations of antiretroviral drugs already affect the propensity of HIV-1-infected individuals infected by M. tuberculosis

10. Lines 107-108: The sentence “There was evidence of a completed course … and 104 (41.6%) in the HIV group.” is not clear to me what the authors attempted to say.

11. Line 118: The authors analyzed SNP frequency using different models, including codominant, dominant, recessive overdominant, and log-additive (Supplementary Table S2). Please provide more details on the difference in each model in Materials and Methods. And why only the dominant model was tested for rs3764880 (TLR8)?

12. Relevant to the previous comment, the authors claimed that rs498670-G (TLR4) (lines 118-119) is critical according to the dominant model. Relative to rs498670-G (A/A) rs498670-G (A/G-G/G) showed few sample numbers according to readouts from the dominant model. In contrast, the sample number of rs5743810 (A/A) is also less than rs5743810 (G/G-A/G), however, rs5743810 (A/A) is defined using the recessive model.

13. Line 121: The authors conclude that rare alleles (rs4986790-G and rs5743810-A) were protective in relation to the risk of developing tuberculosis coinfection. Experimental evidence supporting this conclusion seems weak to me. The authors do observe the correlation; however, the observation cannot fully answer the question of whether or not these rare alleles functionally protect HIV-1-infected individuals from M. tuberculosis infection. For instance, the authors should verify whether the signaling pathways downstream or relevant to these two TLRs and tuberculosis coinfection are also influenced by observed SNPs.

14. Section 2.3. The authors need to provide details in Materials and Methods regarding how they purify CD4 T cells from clinical samples and the purify of purified CD4 T cells.

15. Section 2.3. The whole concept described in Section 2.3. is suspicious to me. T-cell exhaustion (the decline of CD4 T-cell count) is one of the important features alongside HIV-1/AIDS progression no matter whether tuberculosis is present or not. T-cell exhaustion simply reflects a poor state of immunity in HIV-1-infected individuals who can be suffered from many other opportunistic infections. In other words, the measurement of this parameter (T-cell exhaustion) cannot provide us with direct evidence saying that a drop in the number of CD4 T cells harboring SNPs is tuberculosis-specific. The authors should attempt to show that in vitro CD4 T cells harboring indicate SNPs are more permissive to tuberculosis coinfections than infections with other opportunistic pathogens.

16. Discussion should be improved in writing. Several paragraphs seem to summarize current findings but do not link to findings in this manuscript (lines 194-206).

17. Cited reference [39] is relevant to dendritic cells. How much is this paper relevant to the authors’ study focusing on CD4 T cells?

18. Lines 188-189: Reference is required for the sentence “TLRs located on the surface of immune cells … to recognize HIV-1”.

19. Throughout the whole manuscript only the content in Conclusions honestly reflects the contribution of this work based on their findings; other statements/summaries made by the authors seem to be overinterpreted.

Minor comments:

1. Lines 27 and 59: Add a space between two sentences.

2. Line 73: Shi et. all or Shi et. al.?

3. Line 111: Please indicate which LTR harboring rs5743708.

Minor editing is required.

Author Response

Dear reviewer!

Thank you for detailed review of our manuscript. We've made major revisions in our manuscript and try to answer all your questions.

1. Line 33: it is debated whether current ART can be considered “effective” in blocking HIV-1 replication. Indeed, current antiretroviral drugs can block new infections of susceptible cells, but not suppress virus production by cells with integrated proviruses, thus the therapy cannot be interrupted. To be honest, none of the currently available antiretroviral strategies can completely purge HIV-1 infections.

Thank you for your comment. We’ve proved this sentence. 

2. Line 39: Please describe in detail what the personalized approach is and cite the corresponding references.

Thank you for the comments. We’ve completed our manuscript with this data.

3. Line 53: The term “reactivation” is perhaps not a proper word here in this sentence. I went through the reference of Cromarty et al. (2019) [6]. CD4+ T cells that are viral-free should be activated by TLR agonists rather than reactivated.

We absolutely agree with your comment. Replaced the term with the appropriate one.

4. Lines 56-57: Siliciano et al. (2014) [7] is indeed one of the most important milestone papers regarding HIV-1 latency and latent reservoir. However, a lot of progress has been made in the field of HIV-1 latency associated with epigenetic features in the latest 20 years. The authors should provide more updated knowledge at this stage.

Thank you for your comment. We’ve complete the introduction section by some data in this area.

5. Line 56: The same as the previous comment, the sentence “.. resting CD4+ T cells are a basic reservoir of …” is too general. What do the authors mean by “basic reservoir”? In addition, the barrier to curing HIV-1 is due to the persistence of (deeply) latent proviruses in reservoir cells. Resting CD4+ T cells are one type of cells where latent proviruses preferentially integrate; however, those cells are not the reason why we cannot completely purge HIV-1 infection. The key is that we still do not know the location of latent reservoir (what subsets of CD4+ T cells or in which genomic regions for example) harboring (deeply) latent proviruses.

Thank you for your comment. We replaced the term “basic” to term “one of” to make this sentence more clear and entered the information about the lacking data of CD4+ T-cells as a reservoirs to latent proviruses.

6. Lines 59-62: Reference is required for the sentence “Single nucleotide polymorphisms in TLR genes … on the surface of mycobacteria that cause tuberculosis.”.

Thank you, we’ve included corresponding reference.

7. Lines 62-63: Reference is required for the sentence “However, the current data on … with SNPs in TLR genes is contradictory.”.

Thank you, we’ve included reference. 

8. The term “long-term HIV” is ambiguous to me. Is it referred to HIV-1-infected individuals subjected to a long period of ART or a persistent HIV infection? 

In this case, the term "long-term HIV" was used to refer to chronic HIV-infection with tuberculosis coinfection. We've revised it in the manuscript.

9. Section 2.1: If applicable, the authors should provide more information about the treatment history of HIV-1-infected individuals, including the period of ART and what kinds of antiretroviral drugs are used. Can different combinations of antiretroviral drugs already affect the propensity of HIV-1-infected individuals infected by M. tuberculosis? 

Thank you for your comment. Unfortunately, our study did not include the collection of information on ART regimens, as the risk of tuberculosis development is influenced by the virologic and immunologic efficacy of ART, regardless of the regimen received. Despite of some differences in the treatment of HIV-1-infected individuals between the countries, patients were treated according to national guidelines based on World Health Organization recommendations. Thus, most of the patients received a similar ART regimen. Furthermore ART have no effect on genetic factors, opposite of it genetic factors may have some influence to peculiarities of an individual's perception of ART. Studying the effect of different combinations of antiretroviral drugs on the risk of developing opportunistic infections may be a goal for future research.

10. Lines 107-108: The sentence “There was evidence of a completed course … and 104 (41.6%) in the HIV group.” is not clear to me what the authors attempted to say.

Thank you for your comment . To avoid misunderstandings, we have revised the sentence.

11. Line 118: The authors analyzed SNP frequency using different models, including codominant, dominant, recessive overdominant, and log-additive (Supplementary Table S2). Please provide more details on the difference in each model in Materials and Methods. And why only the dominant model was tested for rs3764880 (TLR8)?

Thank You for comment. Since the analysis of genetic associations was carried out in the SNPAssoc package (R environment) using common risk calculation algorithms for different genetic models, we did not include a detailed description of all models, but we make some improved and added the necessary references to the Materials and Methods. 
Regarding the use of the dominant model for the rs3764880 (TLR8). This locus is located on the X-chromosome and, therefore, in men rs3764880 (TLR8) is represented by only one of the two alleles (heterozygotes are impossible). For a sample that includes both men and women, only dominant (major allele homozygotes female + major allele male vs. heterozygotes female + minor allele homozygotes female + minor allele male) or recessive (minor allele homozygotes female + minor allele male vs. major allele homozygotes female + heterozygotes female + major allele male) risk models can be used. Results for recessive model were added to Supplementary Table S2. For both models, we did not obtain a statistically significant association with TB risk in HIV patients.

12. Relevant to the previous comment, the authors claimed that rs498670-G (TLR4) (lines 118-119) is critical according to the dominant model. Relative to rs498670-G (A/A) rs498670-G (A/G-G/G) showed few sample numbers according to readouts from the dominant model. In contrast, the sample number of rs5743810 (A/A) is also less than rs5743810 (G/G-A/G), however, rs5743810 (A/A) is defined using the recessive model.

All genetic risk models were selected based not only p-value but on the AIC and BIC criteria (not provided). The lower this score, the better the model fits the data from which it was created. Analysis of AICs and BICs showed that the most suitable risk model for rare allele G of rs4986790 was dominant, whereas for rare allele A of rs5743810 most appropriate one was recessive model.
In the case of the rs4986790, the allele G exhibits an effect both in the homozygote and in the heterozygote, and in the analysis we combined patients with the GG and AG genotypes into the modified risk group, but the patients with the AA genotype into the common risk group, for which the risk was defined as 1 (non-risk). At the same time, in the case of rs5743810, the A allele exhibits an effect only in the homozygous state and does not manifest itself in the heterozygous one. Therefore, we selected only patients with the AA genotype into the modified risk group, and combined patients with the GG and AG genotype into the common risk group. Modified risk means that a rare allele can act both as high risk allele and as protective allele.

13. Line 121: The authors conclude that rare alleles (rs4986790-G and rs5743810-A) were protective in relation to the risk of developing tuberculosis coinfection. Experimental evidence supporting this conclusion seems weak to me. The authors do observe the correlation; however, the observation cannot fully answer the question of whether or not these rare alleles functionally protect HIV-1-infected individuals from M. tuberculosis infection. For instance, the authors should verify whether the signaling pathways downstream or relevant to these two TLRs and tuberculosis coinfection are also influenced by observed SNPs.

Thank you for your comment. Indeed, genetic factors in the host organism, such as SNP, need to be analyzed in conjunction with a large number of other factors. Most studies on the search for genome-wide associations, unfortunately, cannot answer the question of what the nature of the found associations is and whether risk alleles affect the risk of developing the disease directly or through other mechanisms. Thus, additional research is needed to understand what mechanism is involved in TB in people living with HIV and how the associations we have found can influence the onset of the disease. This not only involves looking for association SNPs but also looking for expression levels and other genetic/epigenetic factors. Our study focused specifically on finding associations of various factors that lead to secondary diseases such as tuberculosis in people living with HIV. 

14. Section 2.3. The authors need to provide details in Materials and Methods regarding how they purify CD4 T cells from clinical samples and the purify of purified CD4 T cells.

Thank you for the suggestion. We’ve provide detailed information about purifying CD4 T cells from clinical samples and protocols of CD4 T-cells count. This part of the study were providing on flow cytofluorimeter FACSCalibur (Becton Dickinson, USA) according to standard manufacturer protocols.

15. Section 2.3. The whole concept described in Section 2.3. is suspicious to me. T-cell exhaustion (the decline of CD4 T-cell count) is one of the important features alongside HIV-1/AIDS progression no matter whether tuberculosis is present or not. T-cell exhaustion simply reflects a poor state of immunity in HIV-1-infected individuals who can be suffered from many other opportunistic infections. In other words, the measurement of this parameter (T-cell exhaustion) cannot provide us with direct evidence saying that a drop in the number of CD4 T cells harboring SNPs is tuberculosis-specific. The authors should attempt to show that in vitro CD4 T cells harboring indicate SNPs are more permissive to tuberculosis coinfections than infections with other opportunistic pathogens.

Thank you for the comment. The declining of CD4 T-cell count is really one of the important features alongside HIV-1/AIDS progression no matter whether tuberculosis is present or not. At the same time tuberculosis remains one of the leading cause of death of people living with HIV. Our findings does not prove that the CD4 T cells containing the risk alleles of TLR genes are more favorable for tuberculosis coinfections, but HIV-1-patients carrying risk alleles have lower CD4 counts regardless of TB coinfection. Thus, a decrease in the number of CD4 T cells carrying risk alleles may not be TB-specific. Verifications of our findings at CD4 cell culture is indeed a very interesting and important work that requires a separate well-planned study, which are beyond the scope of this article. We hope that in the future we will be able to carry out such work.

16. Discussion should be improved in writing. Several paragraphs seem to summarize current findings but do not link to findings in this manuscript (lines 194-206).

Thank you for the comment. We have made major changes to the discussion section.

17. Cited reference [39] is relevant to dendritic cells. How much is this paper relevant to the authors’ study focusing on CD4 T cells?

TLRs are presented both on the surface of dendritic cells and CD4. Thus, Authors hypothesized that modulation of TLR expression may be a mechanism that promotes HIV-1 replication and the pathogenesis of AIDS in patients co-infected with opportunistic pathogens. In addition to increased TLR4 expression in pDCs, its activity might also, thereby, induce increased loss of CD4 + T cells via apoptosis. On the other hand, polymorphisms in TLR genes can also be factors influencing TLR expression. 

18. Lines 188-189: Reference is required for the sentence “TLRs located on the surface of immune cells … to recognize HIV-1”.

Thank you, we’ve completed the reference.

19. Throughout the whole manuscript only the content in Conclusions honestly reflects the contribution of this work based on their findings; other statements/summaries made by the authors seem to be overinterpreted.

Thank you for the comment. In conclusion, we have included only the most important and reliable results from our study. The rest of the results require further confirmation in other similar studies.

Minor comments:

1. Lines 27 and 59: Add a space between two sentences.

Thank you. We’ve corrected this.

2. Line 73: Shi et. all or Shi et. al.?

Thank you for noticing the typo, we’ve corrected it.

3. Line 111: Please indicate which LTR harboring rs5743708.

There were 2 SNP (rs5743708 and rs3804100) which are located in TLR2. We’ve complete this misprint in the text.

Round 2

Reviewer 3 Report

Comments for the Authors

epigenomes-2453853

Association Of Toll-Like Receptors Genes (TLR) Polymorphisms With Tuberculosis In HIV-positive Participants

Authors: Svetlana Salamaikina, Vitaly Korchagin, Ekaterina Kulabukhova, Konstantin Mironov, Vera Zimina, Alexey Kravtchenko, Vasily Akimkin

The authors have answered all my questions point by point. The quality of this revised version of the manuscript has greatly improved. I do not have further comments.

Author Response

Thank you for the review